# Both Plasticizing and Air-Entraining Effect on Cement-Based Material Porosity and Durability

**DOI:** 10.3390/ma15134382

**Published:** 2022-06-21

**Authors:** Aigerim Tolegenova, Gintautas Skripkiunas, Lyudmyla Rishko, Kenzhebek Akmalaiuly

**Affiliations:** 1Department of Construction and Building Materials, Satbayev University, Satpaev str.22a, Almaty 050013, Kazakhstan; kakmalaev@mail.ru; 2Department of Building Materials and Fire Safety, Faculty of Civil Engineering, Vilnius Gediminas Technical University, Sauletekio al.11, LT-10223 Vilnius, Lithuania; 3HeidelbergCement Klaipeda Research Laboratory, Švepelių g. 5, LT-94103 Klaipėda, Lithuania; rlp06@ukr.net

**Keywords:** cement, air-entraining admixture, plasticizing, porosity, air-content, strength, freeze-thawing resistance

## Abstract

The influence of a complex application of both plasticizing and air-entraining effects on concrete with polycarboxylate ether superplasticizer (PCE), air-entraining admixture (AIR), or an anti-foaming agent (AF) is analyzed in this paper with considerations for on the air content, workability, flexural and compressive strength, and freezing–thawing resistance of hardened cement mixtures. The effect of the complex behavior of PCE, AIR, and AF on the porosity of hardened cement mortar (HCM) and freezing–thawing resistance was investigated; freezing–thawing resistance prediction methodology for plasticized mortar was also evaluated. The results presented in the article demonstrate the beneficial influence of entrained air content on consistency and stability of cement mortar, closed porosity, and durability of concrete. Freezing–thawing factor K_F_ depending on porosity parameters can be used for freezing–thawing resistance prediction. With both plasticizing (decrease in the water–cement ratio) and air-entraining effects (increase in the amount of entrained air content), the frost resistance of concrete increases, scaling decreases exponentially, and it is possible to obtain great frost resistance for cement-based material.

## 1. Introduction

Currently, construction, regardless of the purpose of the buildings being created, is characterized by high requirements for the quality of materials. In the construction market, the leading position is occupied by cement concrete. Significant importance today is given to improving the durability of concrete, especially cement compositions. For various operating environments, the durability of concrete is achieved by increasing the entrained air content, reducing the W/C, and increasing the strength class of concrete, as well as cement consumption, using a limited number of types of cement and normalizing their mineralogical composition [1,2,3].

Concrete has the potential to be damaged if it is subjected to freeze–thaw cycles. Therefore, determining how to scientifically optimize the ratio of high-efficiency concrete raw materials and accurately predict the frost resistance of concrete to improve the durability of its application is of great scientific significance. The prediction accuracy of frost resistance is highest when the concrete mix proportion factors considered are the water binder ratio, cement content, fine aggregate dosage, coarse aggregate dosage, and compound superplasticizer dosage [4,5,6]. Seong-Tae [7] suggested a prediction method for minimum curing time based on the hydration of the cement. According to this method, the rate of the decrease of compressive strength increases when the onset time of frost damage is faster and the water–cement ratio is higher. Authors Zeng et.al revealed that graphene oxide can improve frost resistance and compressive strength of air-entrained mortars by 18.9% and 41.9%, respectively [8]. An important technical way to improve the frost durability of concrete is using air-entraining admixtures [9]. Air void structure is the key parameter that affects the frost resistance of concrete [10]. The entrained air void system in cement mortar and concrete directly affects both the fresh-state workability and the freeze–thaw durability of concrete pavements and structures [11,12]. The author Feng Yu [13] investigated the compounding use of AF and VMA which significantly improved the workability, air void structure, and the frost resistance of concrete. According to Feng Yu’s results, at 0.0055% air-entraining agent (AIR) dosage, the combination of 0.15% AF and 0.015% viscosity modifying agent (VMA) reduced the number of large bubbles in concrete by 57.96%. Afterward, the number of fine bubbles was increased by 16.55% and the spacing factor was reduced by 18.09%. A decrease in spacing factor increases freeze–thaw resistance of concrete. 

Air content is an important factor for achieving proper porosity of concrete. Zheng [14] and Rodríguez [15] presented test results that show that frost resistance increases with increasing air content, which makes the space parameter decrease in the hardened concrete. Moreover, the experimental results indicate that, in air-entraining concrete, total air content is not the only factor that affects the final properties of the concrete; air void structure parameters, including void size, shape, and distribution, are key factors as well [16]. The research results of Lazniewska-Piekarczyk et al. [17] on the influence of air-entraining admixture type have proved that a greater amount of micro pores in concrete is the effect of a synthetic air-entraining admixture. The combination effect of air-entraining admixtures on freeze–thaw resistance of cement mortar was also considered in [18].

The porosity of cement mortar is one of the most important microstructural features, which manifests at different length scales [19]. Capillary pores have the biggest influence on the resistance of concrete. Gel pores and water inside them have no influence on the frost resistance of concrete. In fact, water, due to its greater density, does not freeze in those pores. Closed air pores perform the role of reserve pores and increase the frost resistance of concrete [20]. Zhang’s [16] analyses concluded that as the number of freeze–thaw cycles increases, the repeated action of periodic freezing and expansion forces and hydrostatic pressure on the pores inside the concrete cause the pores inside the concrete to gradually expand, penetrate, and form connected pores. Shinichi [21] investigated pore structure in high-strength concrete at early ages via the BSE imaging technique assuming the Powers model. At a very early age, most of the capillary pores in ordinary concrete are so large that their pore size distribution was discontinuous.

Moreover, frost resistance of concrete depends both on open porosity (the amount of capillary pores) and closed porosity (air content in the mixture), and quantitatively can be determined by the frost resistance factor K_F_, which is derived from the equation [22]:(1)$$K_{F}=\frac{P_{c}}{0.09\cdot P_{o}},$$
where $P_{c}$—closed porosity of hardened cement mortar (air pores) and $P_{o}$—open porosity of hardened cement mortar (capillary pores).

The authors Funk [23] and Setzer [24] performed a comprehensive testing program before, during, and after standardized freeze–thaw weathering (CDF). Freeze–thaw cycling caused considerable deteriorations which were significantly modulated by the different admixtures via changes in cement micromorphology. According to the authors of [25,26], the frost resistance of concrete is determined by its porosity because water can only penetrate open pores. Capillary pores have the greatest influence on the conglomerate’s frost resistance. They are open and simple to fill with water [27]. As for the air pores, as opposed to capillary pores, they increase the conglomerate’s frost resistance. During the immersion process the air pores are closed and no water enters them [28].

According to Łazniewska-Piekarczyk [29], the influence of the type of admixtures on porosity and pore size distribution of high-performance self-compacting concrete (HPSCC) at a constant level of water were analyzed whilst considering the cement coefficient, the type and volume of the aggregate, and the volume of the cement mortar. Despite the fact that the air content parameters differed from the standard recommendations, she found that HPSCC was frost resistant. The authors [30] investigated the freeze–thaw behavior of air-entraining cement mortars saturated with a NaCl solution at a concentration of 10 wt.% using experimental measurement approaches. The findings of these studies showed that air-void entrainment tends to decrease thermal contractions but increase hydraulic expansion, ice nucleation expansion, and residual expansion. Some scholars modified cement with polypropylene (PP) fiber, such as Ping Jiang [31], and they found that with an increase of fiber from 0.25% to 1% the porosity was increased by approximately 6% due to the random distribution intersection and interleaving of (PP) fibers in concrete. The goal of this study is to investigate the technological properties and effects of air voids and porosity parameters of cement with different water–cement (W/C) ratios under the action of freezing–thawing cycles. The freeze–thaw (CDF) (test) technique is used to determine the surface scaling of the specimens. Strength of hardened cement mortar and porosity parameters are calculated in relation to freezing–thawing factors. In previous investigations, it has been noted that an important parameter for frost resistance is a decrease in capillary porosity using plasticizers and an increase in the amount of air in concrete using an air-entraining admixture. No research studies have investigated the combined effect of plasticizing and air-entraining admixtures on the freezing–thawing resistance of hardened cement.

## 2. Materials and Methods

### 2.1. Materials

#### 2.1.1. Cement

Portland cement without mineral admixtures CEM I 42.5 R conforming to EN 197-1 with water consumption of 26.6% was used. The mineral composition, physical properties, and mechanical properties of the cement are presented in Table 1, Table 2, and Table 3, respectively.

The results of the particle size distribution of cement are presented in Figure 1.

#### 2.1.2. Aggregate and Water

Natural river sand with a fraction of 0/2 was selected as fine aggregate. The grain-size distribution for the sand is shown in Figure 2.

Potable water was used for concrete mixtures. Water confirms the requirements of EN 1008.

#### 2.1.3. Polycarboxylate Ether Superplasticizer (PCE)

Polycarboxylate ether is used as the superplasticizer admixture to improve the workability of cement mortar. The physical and chemical properties of PCE are shown in Table 4.

#### 2.1.4. Air-Entraining Admixture (AIR)

SikaControl-10LPSA is a synthetic surfactant based on a brownish concrete admixture in liquid form. The physical and chemical properties of SikaControl-10LPSA are shown in Table 5.

#### 2.1.5. Antifoaming Agent (AF)

The polypropylene ether was used as a component that prevents foaming of the mortar. Antifoam is a chemical agent used to reduce and prevent the formation of foams during chemical mixing. Foams will cause serious problems in chemical processes and will also prevent utilization of the whole capacity of a container. Developing conveniently and rapidly on the surface of the foam is one of the main features of antifoams [32]. The physical and chemical properties of AF (A-316) are shown in Table 6.

#### 2.1.6. Mix Proportion

The composition, consistency (determined according to EN 1015-3), and density (determined according to EN 1015-6) of fresh cement mixtures are presented in Table 7 and Table 8. The first five specimens have been mixed with a polycarboxylate ether and designated as PCE; the other three specimens have been mixed with an air-entraining admixture. In superplasticizer mortars, the main difference between mixtures is the different W/C ratio, which varies from 0.50 and 0.38. When changing the W/C ratio, the amount of cement and water remained the same. In the case of mortar with an air-entraining admixture, the W/C ratio was equal to 0.4 for all specimens.

A Portland cement sample (sample C0) was prepared for the reference sample. Three chemical admixtures were used to create seven composite admixtures with the same mass ratio as follows: polycarboxylate ether in solid-state (symbol PCE), Anti-foaming admixture (symbol AF), and Air entraining admixture SikaControl-10LPSA (symbol AIR).

Variations of polycarboxylate ether content by mass were 0%, 0.1%, 0.2%, 0.3%, and 0.4%, whereas variations of Air entraining admixture Sika (AIR) content were 0.1%, 0.2%, and 0.3% for total cementitious materials. The control samples without admixture were prepared.

Dosage of anti-foaming admixture was constant with all specimens (10% from superplasticizer content).

Table 7 shows that with the subsequent increase in admixture (PCE and AF) in each receipt there was a decrease in the amount of water from 100% to 77%.

### 2.2. Methods

#### 2.2.1. Mixing Procedure and Mixture Properties of Cement Mortar

Cement mortar components were mixed according to the EN 196-1 procedure.

The proportions by weight were one part cement, three parts of 0/2 sand, and a water/cement ratio of 0.50.

Polycarboxylate ether and air entraining admixtures were stirred with the mixing water at high speed for 1 min to obtain uniform dispersion. 

After mixing of components, the consistency of fresh mortar was determined by a flow table method (EN 1015-3). The flow value was determined by measuring the spread diameter of the test samples.

The bulk density of fresh mortar was standardized by weighing a known volume of fresh mortar according to EN 1015-6, using a sample of approximately 200 mL of fresh mortar for each composition.

#### 2.2.2. Properties of Hardened Cement Mortar

The test specimens were 40 mm × 40 mm × 160 mm^3^ prisms and 100 mm × 100 mm × 100 mm^3^ cubes.

Cubes were molded for determining durability; prisms were molded for density, flexural and compressive strength, and porosity testing.

The density and linear dimensions of cement prisms and cubes were tested according to EN 1015-1.

The specimens were horizontally immersed in water at a temperature of (20.0 ± 1.0) °C in containers for curing.

After 14 days, tests of the mortars were carried out for flexural and compressive strength according to EN 196-1. The specimens were loaded using a flexural and compression strength testing machine UTEST UTCM-1100.

#### 2.2.3. The Porosity of Hardened Cement Mortar

The porosity parameters of hardened cement mortar were determined by measuring kinetics of water absorption. According to this methodology, open porosity (capillary pores), total porosity, and closed porosity (air pores) of the hardened cement mortar are defined.

The test of samples with dimensions about 40 mm × 40 mm × 40 mm^3^ after splitting was carried out in a dried state to a constant mass. The samples were placed in a container filled with water so that the water level in the container was about 50 mm higher than the upper level of the stacked samples. The water temperature in the tank was (20 ± 2) °C.

The samples are weighed 15, 30 min, and 4 h after immersion of the dried sample in water, and then every 24 h to a constant weight. According to the test results, the relative water absorption by mass was calculated. The porosity and parameters of the series of hardened cement samples are determined as the arithmetic mean of the test results of four samples of the series.

The total, open, and closed porosity of hardened cement mortar was calculated after determination of water absorption kinetics. The total porosity of hardened cement mortar is calculated by equation:(2)$$P_{b\;}=\left(1\;-\frac{\rho_{b}}{\rho_{s}}\right),$$
where $P_{b}$—is the total porosity of hardened cement mortar; $\rho_{s}$—is the specific density of cement mortar, 2690 kg/m^3^; and $\rho_{b}$—is the density of hardened cement mortar.

Open porosity (capillary pores) of hardened cement mortar is calculated by equation:(3)$$P_{a}{=W}_{p}\cdot \frac{\rho_{b}}{1000},$$
where $P_{a}$—is open porosity of hardened cement mortar (capillary porosity), %; and $W_{p}$—is water absorption of hardened cement mortar, %.

The closed porosity of hardened cement mortar (air porosity) is calculated as the difference between the total porosity and the open porosity.

The density of cement mortar specimens used for water absorption testing was measured according to EN 12390-7, and the volume of a specimen was determined by the water displacement method.

#### 2.2.4. Freezing–Thawing Test Procedure

The freezing–thawing resistance of cement mortar samples was provided in the following way. The cubic cement mortar specimens after 7 days of curing were removed from the water bath and sawn in two parts perpendicular to the top surface. The test surface of the cement mortar specimen is the side surface of the cube from the mold (not the cut surface of the cube). The prepared specimens were put into the container by testing the surface of the cube downwards, supported by gaskets, and were immersed in 3% NaCl water solution to a depth of about 5 mm for 7 days before placement in the cooling chamber. The testing scheme for freezing–thawing of cement mortar specimens is presented in Figure 3. The freezing–thawing test was performed in a 3% solution of sodium chloride freezing medium. The specimens after saturation were placed into the cooling chamber and subjected to repeated freezing and thawing according to a time–temperature cycle from 20 ± 4 °C to 20 ± 2 °C and backward to 20 ± 4 °C for 24 h.

The collection of scaled materials from cement mortar specimen test surfaces was performed after 7, 14, 21, and 28 cycles. The tested specimens’ surface was rinsed and brushed to remove the scaled materials into the special vessel until no further scaled material could be removed. The liquid with scaled material from the container was poured carefully through a filter paper. The collected materials in the filter paper were washed to remove any remaining sodium chloride. After that, the filter paper and collected materials were dried to constant mass at (65 ± 5) °C for 24 h and weighed to the nearest 0.1 g. After that, the cumulative mass of the dried scaled material and scaling value from the test surface area after n freeze–thaw cycles were calculated.

The total amount of scaled materials related to the test after *n*th cycles m_n_ was calculated for each measuring occasion and each specimen is shown in equation:(4)$$m_{n}=\frac{{\Sigma m}_{s}}{A},$$
where $m_{s}$—is the mass of scaled material of measurement after *n* cycles (kg/m^2^) with an accuracy of 0.01 g. The sum is taken over all measurements until the *n*ts; *A*—is the area of the test surface, m^2^. It is calculated based on specimen linear dimensions. They are taken as the average of at least two measurements determined to the nearest 0.5 mm.

The amount of scaling has been determined after 7, 14, 21, and 28 freeze–thaw cycles. 

## 3. Results and Discussion

Cement mortars prepared with different admixtures at various ratios were examined by the flow table test and some mechanical tests to determine strength, flow, and freezing–thawing resistance. Pore structure development of mortars was investigated. Additionally, 0–0.4% dosages for superplasticizing and air-entraining admixture were used to indicate the admixture base effect.

### 3.1. Consistency, Stability, and Water Requirements

The addition of a certain amount of admixtures caused a decrease in the amount of water demand. At the initial stage, the water-reducing effect of the introduction of polycarboxylate ether into the cement-sand mixture was considered.

Figure 4, obtained by experimental work, shows the influence of PCE by weight on water demand reduction. It was observed that with an increase in the amount of superplasticizer admixture from 0.1% to 0.2% of the cement weight, the decrease in water demand for the modified cement mortars of homogeneous viscosity steadily decreases from 9.8% to 18.1%. With a further increase in the amount of superplasticizer from 0.2% to 0.4%, the decrease in the water demand of the cement mortar increases slightly from 18.1% to 22.8%. The observed phenomenon can be explained by the fact that the effect of water reduction in the availability of a superplasticizer is related to its dispersing ability, expressed through the ζ-potential. Superplasticizers, by reducing the ultimate shear stress and plastic viscosity of the mortar, can increase the characteristics of the fresh cement mortar.

Chemical admixtures are thought to improve flow performance in cement mortar by dispersing cement flocculation. Moreover, the compatibility between cement and chemical admixtures strongly depends on the physical (surface texture characteristics) and chemical (surface charge) characteristics of the additions. The effect of polycarboxylate ether and air–entraining admixture cement mortars on flow table tests can be seen in Figure 5. Figure 5 shows the variation in average flow spread of fresh mortar mixtures prepared with four dosages of PCE and three dosages of AIR in comparison to the control mixture. The flow diameter of the AIR mixture was recorded to be 197 mm, which was expected to be the highest as compared to the PCE-based mixtures. On the other hand, mixtures prepared with PCE exhibited steady diameters from 133 mm to 156 mm, owing to the use of a polycarboxylate ether admixture. Comparing the effects of PCE, it can be found that, when the dosage is less than 0.3%, PCE has no significant effect on fluidity. The flow table test results were similar to the experimental conclusions of Zhang et al. [33].

It was observed that a combination of a PCE, an AIR, and an AF admixture has been shown to increase dough flow better than the combined use of a polycarboxylate ether and an antifoam agent.

The mortar was modified with a different amount of air-entraining admixture (from 0.1% to 0.3% by weight of cement).

Results on relationships between air content and air-entraining admixture are given in Figure 6. It has been indicated that the air content of fresh cement increases with the amount of AIR in a linear fashion.

The efficiency of AIR content increases linearly, from 1.09% to 15.03%, with an increase in the amount of air involved in the mortar, from 0.1% to 0.3% by weight of cement (Figure 6). The stabilizing effect of air-entraining admixtures is ensured by their adsorption on the surface of air bubbles. On adding an air-entraining surfactant to cement mortar, its molecules are inserted between adjacent molecules at the water surface; the mutual attraction between the separated water molecules is reduced. Lowering the surface tension stabilizes the bubbles against mechanical deformation and rupture, making it easier for bubbles to be formed [34].

Effects of air content and type of air-entraining agent on the flowability increment of fresh cement mortar are shown in Figure 7. It can be seen that the flowability increment of fresh cement mortar increases with air content. As expected, it can be clearly seen from Figure 7 present that AIR, irrespective of type, can improve the flow of cement mortar. However, the magnitude of this improvement in fluidity depends on the type and dosage of AIR. The photos show that water separation of the cement and segregation of cement spread does not occur. When the dosage of AIR increased from 0 to 0.1%, the flow spread of mortar increased from 147 mm to 171 mm, and the flow spread increased by 16.3%. At a dosage of AIR from 0.1% to 0.3%, there was an increase in spread flow from 171 mm to 197 mm. Compared to the control sample, the maximum value of fluidity increment, 34%, was achieved by increasing the amount of air-entraining admixture from 0 to 0.3%.

The workability of fresh cement mortar is significantly improved due to the generation of a lot of micro-air bubbles when an air-entraining agent with high quality is added, i.e., its flowability is enhanced; however, its bleeding and segregating capacity are reduced and the cohesiveness and homogeneity of cement mortar are therefore enhanced [35].

In Figure 8, it can be seen that the efficiency of air-entraining agent (AIR) plasticization depends on its content in the Portland cement mortar. It has been established that with an increase in the amount of air-entraining agent (AIR) from 0.1% to 0.3% by weight of cement, the decrease in water demand of cement mortar of uniform viscosity steadily increases from 0 to 7.7%. With a further increase in the amount of air-entraining agent (AIR) from 0.2% to 0.3%, the water demand reduction of mortar slightly decreases from 7.7% to 8.9%. The increase in air content occurred due to improvement of the fluidity of cement mortar. Reducing the amount of water caused the formation of a structure with the smallest number of pores, this fact is confirmed by the results obtained.

### 3.2. Density and Strength of Hardened Cement Mortar Samples

The effect of polycarboxylate ether on the workability of a cement mixture and water consumption causes changes in the density of cement in a fresh and hardened state. It has been established that with an increase in the amount of polycarboxylate ether (PCE) from 0 to 0.2% by weight of cement, the density of modified cement mortar density remains constant. Figure 9 shows that with a further increase in the amount of polycarboxylate ether (PCE) from 0.2% to 0.4%, the density of cement does not change and remains at a constant density of about 2280 kg/m^3^.

Air content is an important factor which affects aspects of cement mortar workability, density, and compressive strength. A comparison of densities of cement mortar modified by an air-entraining agent (AIR) is present in Figure 10. It has been demonstrated that, with an increase of air entrainment (AIR) and air content from 1% to 15%, density linearly decreases from about 2280 kg/m^3^ to 1975 kg/m^3^ with an increase in air content. It is found that density decreases by about 21.8 kg/m^3^ when the air content is increased by about 1%.

Compressive strength (*F_c_*) and flexural strength (*F_f_*) values of cement mortar at an age of 14 days were calculated on the 40 × 40 × 160 mm prisms. Figure 11 shows the influence of the water to cement ratio on the compressive and flexural strength of hardened mortar at 14 days. It can be seen that, with the decrease in water and cement ration from 0.5 to 0.39, both *F_c_* and *F_f_* increased linearly from 74.0 MPa to 102.6 MPa and 7.2 MPa to 8.6 MPa, respectively. A more significant increase was observed in flexural strength. The obtained compressive strength and flexural strength in this article are consistent with the results of Erdem et al. [36].

The average experiment results are listed in Figure 12, with different entrained air content of *F_c_* and *F_f_*, respectively. It was observed that both *F_c_* and *F_f_* increased continuously with the decrease in air content. At the age of 14 days, the compressive strength of hardened cement mortar with 1% air content was about 75% more than cement mortar with 15% air content. The flexural strength of hardened cement mortar was about 72% more than cement mortar with 15% air content. It is noticed that compressive and flexural strength decreases by increasing the amount entrained air, whereas lower strength is caused if more voids exist in the cement mortar. This decrease can be compensated by a decrease in the water–cement ratio due to the plasticizing effect of an air-entraining admixture.

One main shortcoming of air-entraining agents is that the compressive strength of concrete decreases with the increase in the air content [37]. Generally, the compressive strength loss per air content is about 4 ± 6%. Results on the effects of compressive strength loss per air content are displayed in Figure 12. They indicates that at the same cement content and different W/C, the compressive strength loss per air increases with the air content of the hardened cement mortar [38]. This compressive strength loss must be evaluated during the concrete mixture design process. According to impact analyses of air-entraining and superplasticizing admixtures in concrete, Nowak-Michta [39] observed that PCE increased compressive strength by 13–14%, and AIR resulted in a decrease in strength from 5% to 17%.

However, the workability of cement mortar is improved with the increase in air content; therefore, water content can be reduced, i.e., the W/C ratio of concrete with air-entraining agents can be lower. Results in Figure 12 conform this conclusion. The compressive strength of hardened cement mortar with AIR and equal workability is not reduced when the air content grows up to 4–5%.

### 3.3. Porosity Parameters

Results in Figure 13 show that, with an increase in the water–cement ratio from 0.39 to 0.50, the water absorption of cement mortar increases from 4.5% to 6.4%, at which a stable linear increase is observed at the initial stage. Up to 6% water absorption and assumed good concrete durability is obtained when the W/C ratio is up to 0.47. It is suggested that this process was influenced by the increased capillary porosity of hardened cement mortar with increased W/C ratio.

Figure 14 gives results on the relationship between air content in hardened cement mortar and water absorption. It is shown that air content in hardened cement mortar (Y) increases water absorption (X) in a linear fashion:Y = 4.4287 × e^0.0205x^(5)

The variation of the open and closed porosity of the cement mortar depending on the amount of PCE plasticizing admixture is shown in Figure 15. The curves in this figure show that increasing the amount of PCE plasticizing admixture to 0.4% by weight of the cement (P4 composition) decreases the open porosity (capillary) of the cement mortar by about 29% with the reduction of W/C ratio from 0.50 to 0.38, and the closed porosity (air content in the cement mortar) increases from about 1% to about 4% compared to the control mixture without superplasticizer. The following factors can explain this phenomenon. A similar phenomenon is investigated in the work of Zhao et.al [40]. On one hand, open porosity decreased with water content in cement mortar decreasing, and more air bubbles combined with the decrease in the water content and W/C ratio of the cement mortar, thus reducing total bubble surface area. As a result of the increased cement mortar on the bubble surface, the pore wall thickened, as evidenced by decreased open porosity and increased closed porosity. The results presented in publication [41] proved that, with the introduction of air-entrapping admixture, cement mortar on the surface of the bubbles increases and the pore wall thickens accordingly, which manifests in a decrease in open porosity and an increase in closed porosity.

Figure 16 shows that by increasing the amount of air-entraining admixture to 0.3% by weight of cement, both open and closed, the porosity of the hardened cement mortar changed. Open porosity of hardened cement mortar ranges from 10.68% to 11.95% and closed from 4.16% to 14.47%. Compared to the initial porosity of the cement mortar without air-entraining admixture, open porosity is increased by 11.46% and closed porosity is increased by about 71.25%. Open pores and capillaries are formed by the removal of free water, and the number and size of such pores mainly depends on the ratio of water to cement. Closed pores are formed by the incorporation of air into the mixture and the contraction of the hardening cement mortar.

### 3.4. Freezing–Thawing Resistance

The final stage of the study was to determine the freezing–thawing resistance of the samples by the second accelerated CDF method under conditions of pre-saturation of the samples in a 3% NaCl solution. Samples that hardened in air-humid conditions for 7, 14, and 28 days were exposed to the test. 

The surface deterioration of the specimens, modified by carboxylate ether that underwent 14 cycles of freeze–thaw, is shown in Figure 17.

The surface deterioration of the air-entraining hardened cement mortar (HCM) specimens during 14 cycles of freeze–thaw are shown in Figure 18.

The figures show the appearances of specimens after different freeze–thaw cycles. The surface of cubic specimens became rough, and the surface mortar was loosened gradually with the increasing freezing–thawing cycles. With seven freeze–thaw cycles, only slight scaling of the cement mortar was observed on the surface of the specimen. It can be seen that the cement mortar was spalled at the corner of a cubic specimen with polycarboxylate ether admixture after 14 cycles. In the case of samples with additional an air-entraining admixture, no spalls were observed at the corners of the cubes. After 14 cycles, the shape was complete due to the periodic freezing and thawing process, and the mortar around the sample did not fall off seriously. Obtained test results indicated by Nowak-Michta et.al [42] show good scaling resistance results after freeze–thaw cycles.

The freezing–thawing resistance of hardened cement mortar modified by plasticizers in a water freezing medium improved with an increase in plasticizer dosage and reduction of the W/C ratio. Figure 19 presents the scaling of cement specimens with carboxylate polymer dosage from 0 to 0.4% during freezing-thawing cycles.

Figure 20 presents the scaling of cement specimens with air-entraining dosage after freezing–thawing cycles. The maximum scaling was observed for ordinary cement without air-entraining admixture. After 18 freezing–thawing cycles, it exceeded 1 kg/m^2^. The figure clearly shows the advantage of formulations with complex admixtures over the plasticizing admixture composition. As shown in these figures, the scaling rate of both cement specimens increased over the whole process of freeze–thaw cycles. For the air-entraining group, the scaling of specimens increased slightly before 14 cycles. This is attributed to the cement mortar’s capillary pore low water absorption and closed porosity volume compensation effect. As the freezing and thawing cycles increased, the internal micro-cracks and pores of the concrete gradually expanded and became connected, causing the surface of the specimen to crumble. For the polycarboxylate ether group, the scaling rate of cement cube specimens significantly changed before 14 freeze–thaw cycles. However, after 21 cycles, the scaling rate of specimens had an obviously larger increase than the air-entraining group. At 28 freeze–thaw cycles, the scaling rates of cement specimens with polycarboxylate ether were 15.57 kg/m^2^, 13.51 kg/m^2^, 6.61 kg/m^2^, 4.29 kg/m^2^, and 6.87 kg/m^2,^ and with air-entraining admixture were 6.61 kg/m^2^, 2.62 kg/m^2^, 0.97 kg/m^2^, and 0.45 kg/m^2^, respectively.

Furthermore, the relationship between scaling rate and freeze–thaw cycles clearly shows two distinct stages. In the first 0–7 freeze–thaw cycles, the weight of concrete specimens shows a slightly increasing trend or is unchanged during the damage accumulation stage. However, after 14 freeze–thaw cycles, the scaling rate for the PCE and AIR group specimens increases significantly, indicating the damage acceleration stage.

The density and porosity properties of cement mortar after 28 days of curing in normal conditions are presented in Table 9. The data presented in Table 9 show that the highest criterion for the resistance of a hardened cement paste to frost and the predicted resistance to frost according to the number of freezing and thawing cycles are typical for the control sample (C0 composition). The obtained scaling results of cement mortar specimens in this article are consistent with the results of Zheng et al. [8], Xu et.al [43], and Yuan et.al [44]. Increasing the amount of plasticizing admixture from 0.1% to 0.4% by weight of cement decreases the criteria for frost resistance and the predicted frost resistance according to the number of freezing and thawing cycles. The porosity parameters of cement were determined according to the kinetics of water absorption. 

The frost resistance of cement mortar of concrete can be predicted according to the frost resistance factor K_F_. The factor shows that the frost resistance of the tested hardened cement must be about 28 freezing–thawing cycles. It was experimentally determined that after 28 freezing–thawing cycles the scaling of cement increased from 0.45 kg/m^2^ to 15.57 kg/m^2^.

In general, all concrete samples have shown a reduction in mass during the test, and this reduction increases with the increasing number of applied cycles.

Statistical processing of test results by using an exponential function model produced a function (Figure 21) of scaling (m_c_) and a frost resistance factor (K_F_). The correlation coefficient of the function is 0.9246. The test results have shown that frost resistance of cement-based material depends on closed and open porosity, and that parameters can be used for frost resistance prediction. Figure 21 shows that, due to the plasticizing effect of the cement mixture, the scaling ranges from about 15.5 kg/m^2^ to 4.5 kg/m^2^ can be reduced, and with additional air-entraining effect reduction in scaling from about 4.5 kg/m^2^ to 0.45 kg/m^2^ can be achieved.

## 4. Conclusions

According to the experimental results, the following conclusions can be drawn:Due to the formation of the appropriate structure of capillary pores in the hardened cement mortar, the durability of cement-based material depends on the W/C ratio. The W/C ratio can be reduced by using plasticizers and increasing the amount of air content in the mixture. An increase in the amount of carboxylate ether from 0 to 0.4% (in dry material) caused a decrease in water demand for constant workability concrete up to 22.8%, and a reduction of scaling from 15.6 kg/m^2^ to 4.3 kg/m^2^.By increasing the amount of air content in the cement-based material from 1.09% to 15.03%, the consistency of the cement mortar was improved about 34% and a decrease in water demand can be obtained up to 8.9% for the same consistency. The use of air-entraining admixtures significantly increased the fluidity of the cement mortar and increased the stability of air-entrained mixtures.An increase in the amount of air content in a cement-based material from 1.09% to 15.03% leads to a decrease in compressive strength by 42.0% and flexural strength by 39.1%: about 3.0% and 2.8% accordingly for 1% of entrained air. The density of cement mortar in fresh and hardened states using polycarboxylate ethers was constant instead of an air-entraining admixture. The air-entraining admixture reduces hardened cement mortar density from 2280 kg/m^3^ to 1975 kg/m^3^.The combined effect of plasticizing and air-entraining admixtures leads to a negligible increase (about 1%) in open porosity, an increase in closed porosity up to 10%, and a large increase in frost resistance factor (K_F_) (from 0.55 to 13.45) and the frost resistance of concrete (reduction of scaling from 15.57 kg/m^2^ to 0.45 kg/m^2^).By the plasticizing effect reducing W/C ratio from 0.50 to 0.39, an improvement in freezing–thawing resistance can be achieved up to about 4.5 kg/m^2^ scaling. With a joint decrease in the water–cement ratio and an increase in the amount of entrained air content, the frost resistance of concrete increases, the scaling decreases exponentially up to about 0.45 kg/m^2^ scaling, and it is possible to obtain greater frost resistance of cement-based material.

To sum up, the obtained results on investigated cement mortars can be used for selection of freezing–thawing resistance concrete composition. The use of plasticizing and air-entraining admixture positively affects mechanical properties and porosity parameters of cement mortar.

From a practical standpoint, the research results presented in this paper allow for the prediction of freezing–thawing resistance of concrete during the design of the composition of the concrete mixture. The prediction of freezing–thawing resistance can be used for concrete with polycarboxylate ether superplasticizers. The freezing–thawing resistance of concrete will be evaluated in scaling resistance (in kg/m^2^).

## Figures and Tables

**Figure 1 materials-15-04382-f001:**
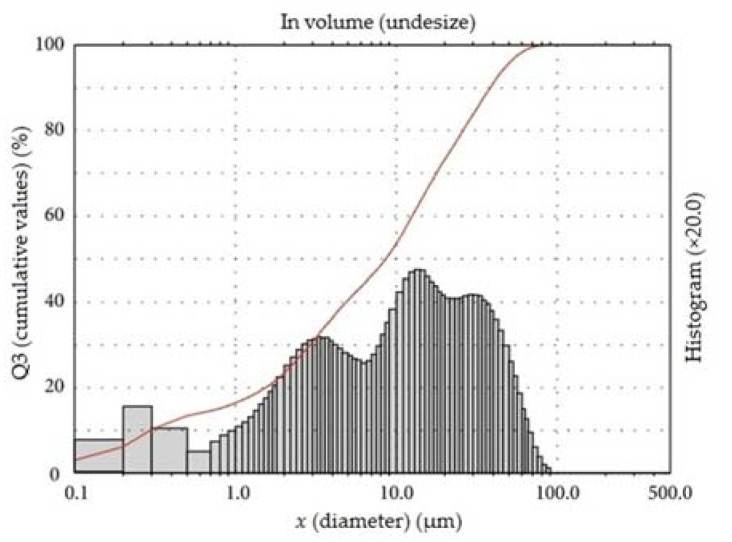
Particle size distribution of cement CEM I 42.5 R.

**Figure 2 materials-15-04382-f002:**
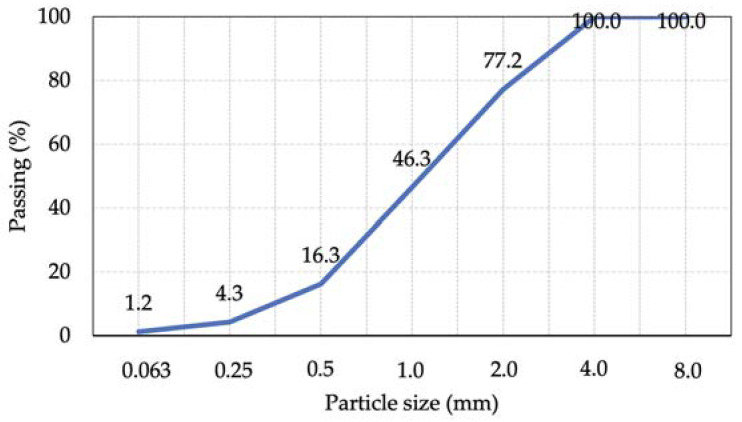
Gradation of fine aggregate.

**Figure 3 materials-15-04382-f003:**
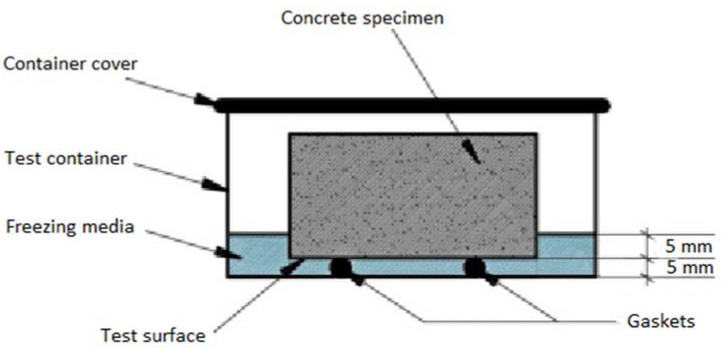
Testing scheme of freezing–thawing resistance of concrete.

**Figure 4 materials-15-04382-f004:**
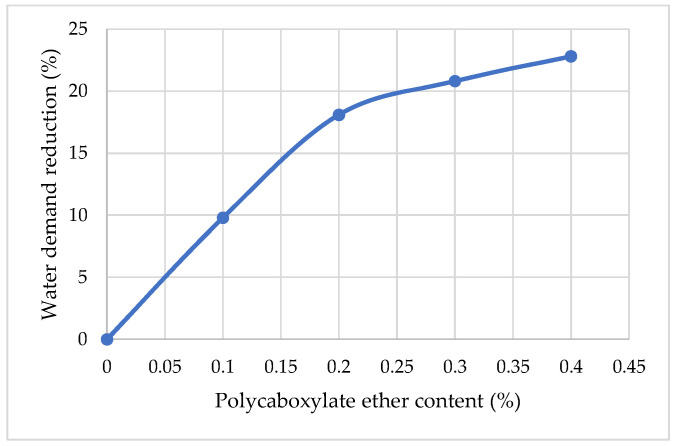
The influence of polycarboxylate ether content on the water demand of cement mortar.

**Figure 5 materials-15-04382-f005:**
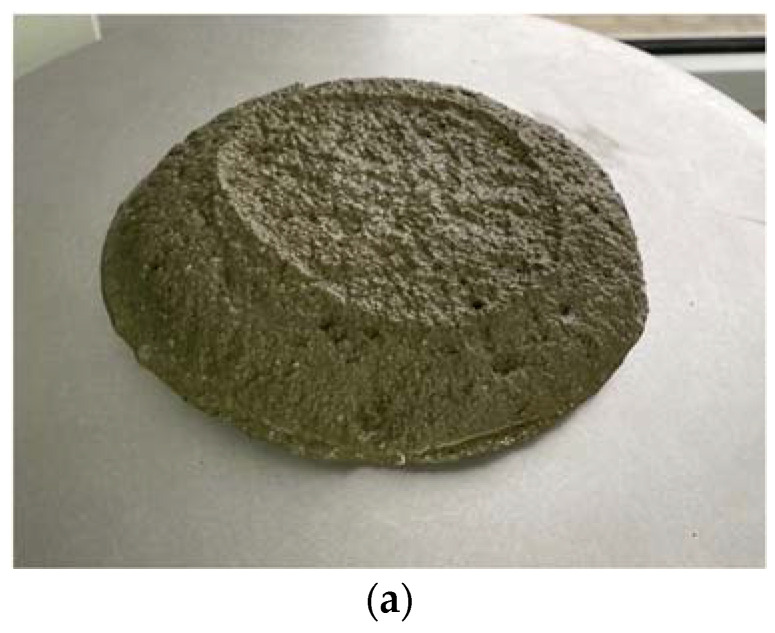
Flow table test result: (**a**) Control specimen; (**b**) Specimen with plasticizer content; (**c**) Specimen with plasticizer and air-entraining content.

**Figure 6 materials-15-04382-f006:**
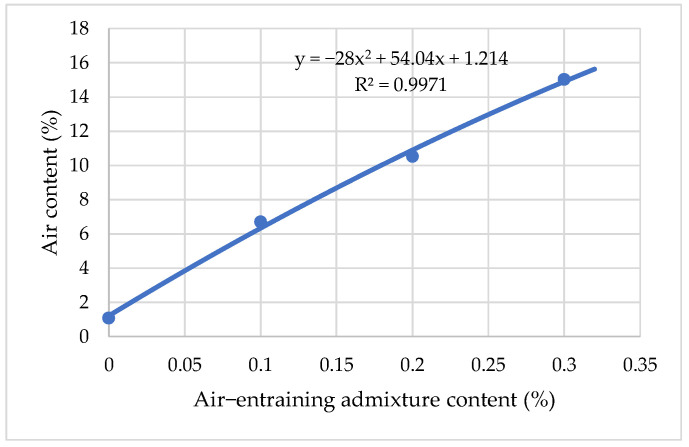
The influence of air-entraining admixture content on air content in cement mortar.

**Figure 7 materials-15-04382-f007:**
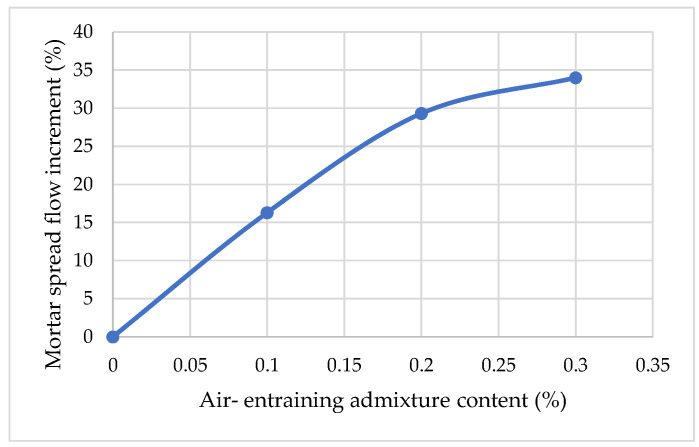
Effect of air-entraining agent on the fresh mortar spread flow increment.

**Figure 8 materials-15-04382-f008:**
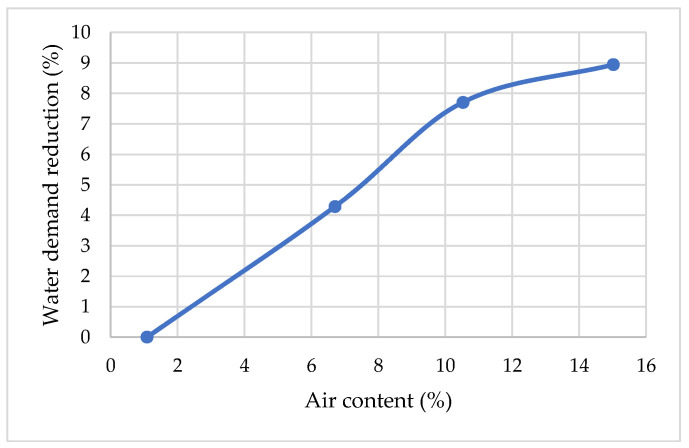
Effect of air-entraining admixture (AIR) on the changes in water demand of modified Portland cement.

**Figure 9 materials-15-04382-f009:**
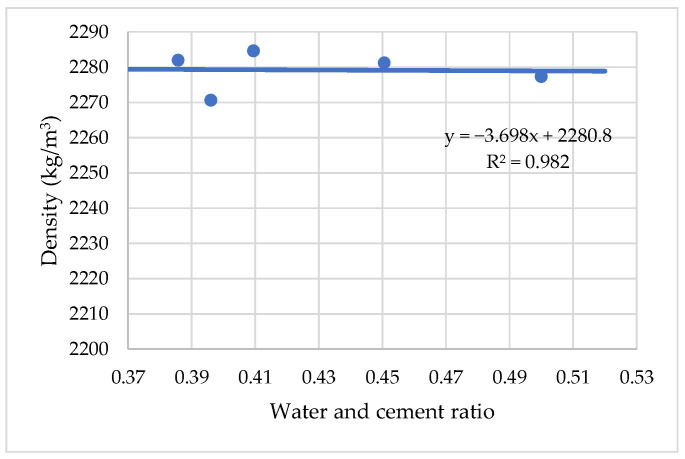
The influence of water cement ratio on density with PCE superplasticizer.

**Figure 10 materials-15-04382-f010:**
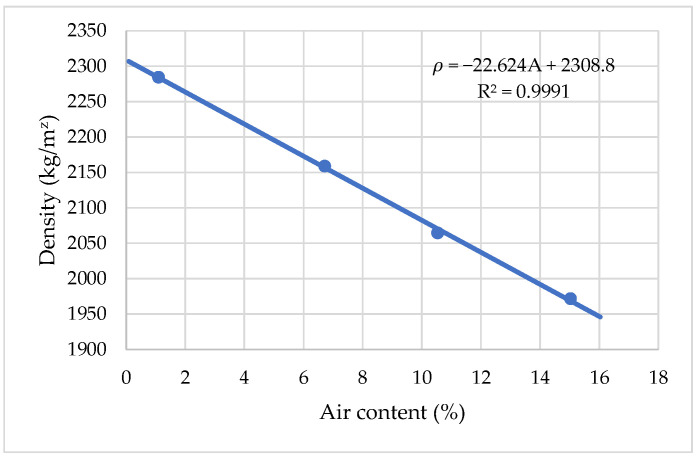
The influence of air-entraining content on cement mortar density.

**Figure 11 materials-15-04382-f011:**
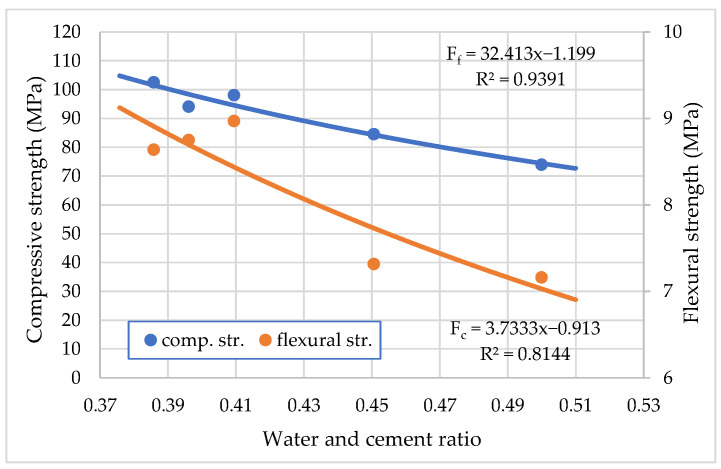
Effect of water to cement ratio on compressive and flexural strength of hardened mortar at 14 days with polycarboxylate ether.

**Figure 12 materials-15-04382-f012:**
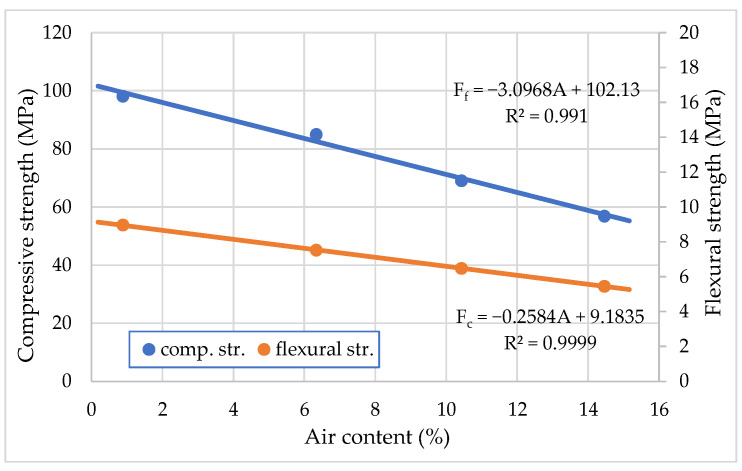
Effect of air content on compressive and flexural strength of hardened mortar at 14 days with an air-entraining admixture.

**Figure 13 materials-15-04382-f013:**
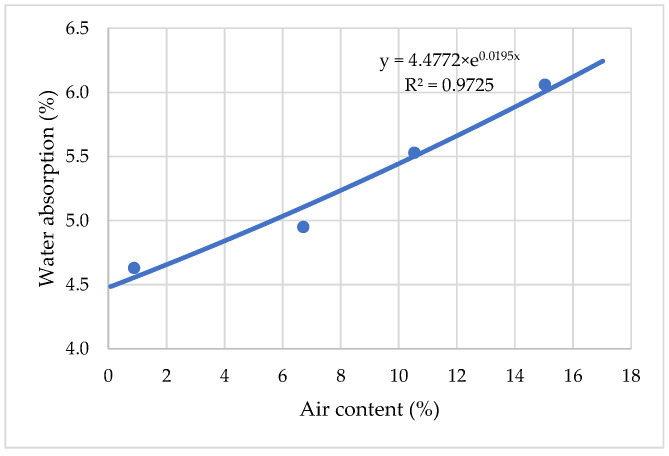
The influence of water absorption on water and cement ratio.

**Figure 14 materials-15-04382-f014:**
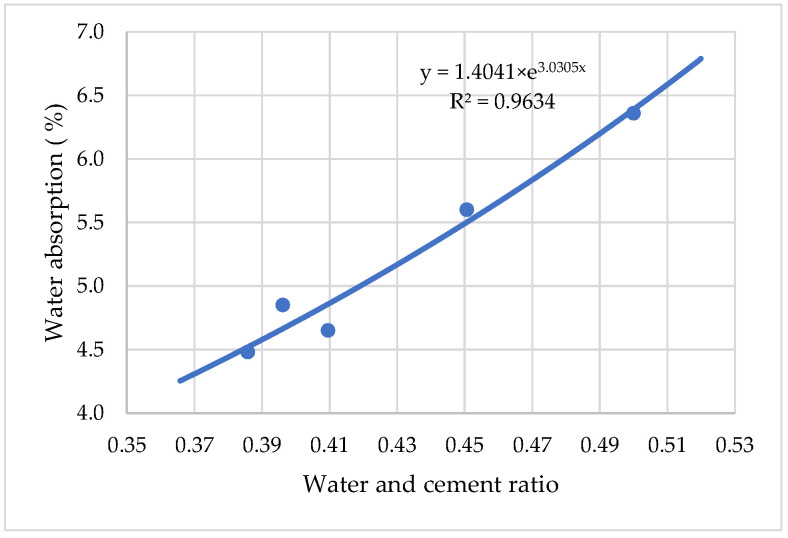
The influence of water consumption on the amount of air-entraining admixture.

**Figure 15 materials-15-04382-f015:**
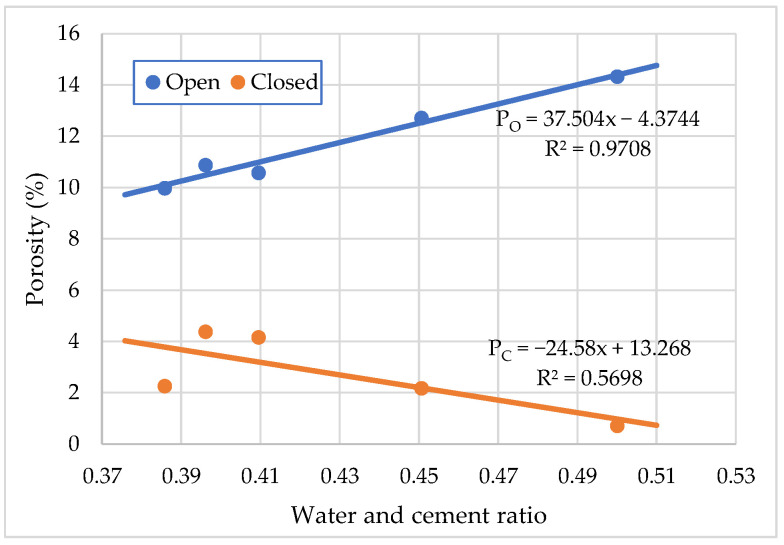
Effect of water to cement ratio on closed and open porosity of cement mortar.

**Figure 16 materials-15-04382-f016:**
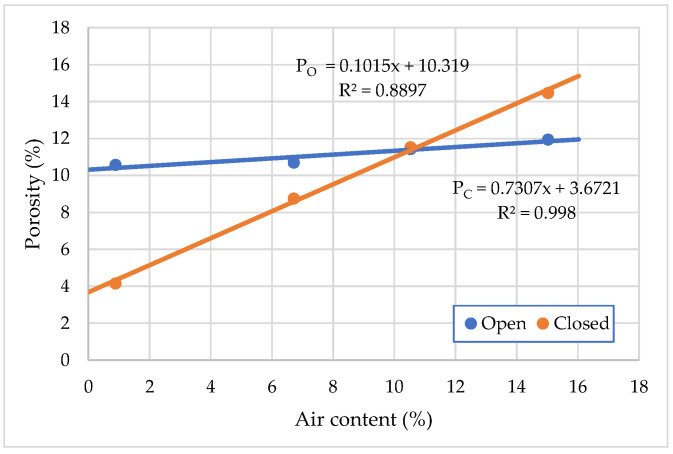
Effect of open and closed porosity changes on air-entraining hardened cement.

**Figure 17 materials-15-04382-f017:**
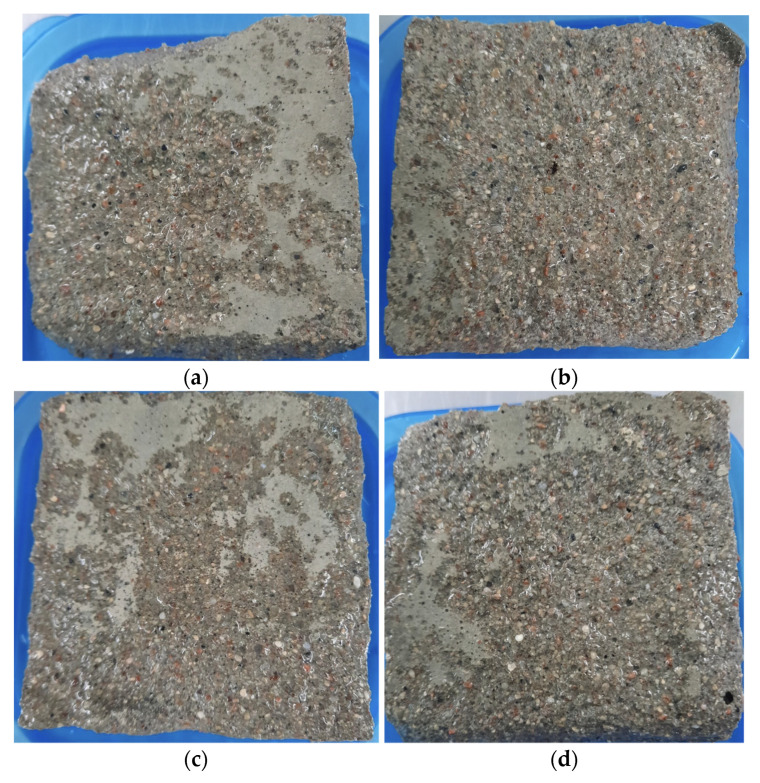
The surface of hardened cement mortar is modified by carboxylate polymer after 14 cycles of freeze–thaw: (**a**) C0 sample; (**b**) P1 sample; (**c**) P3 sample; (**d**) P4 sample.

**Figure 18 materials-15-04382-f018:**
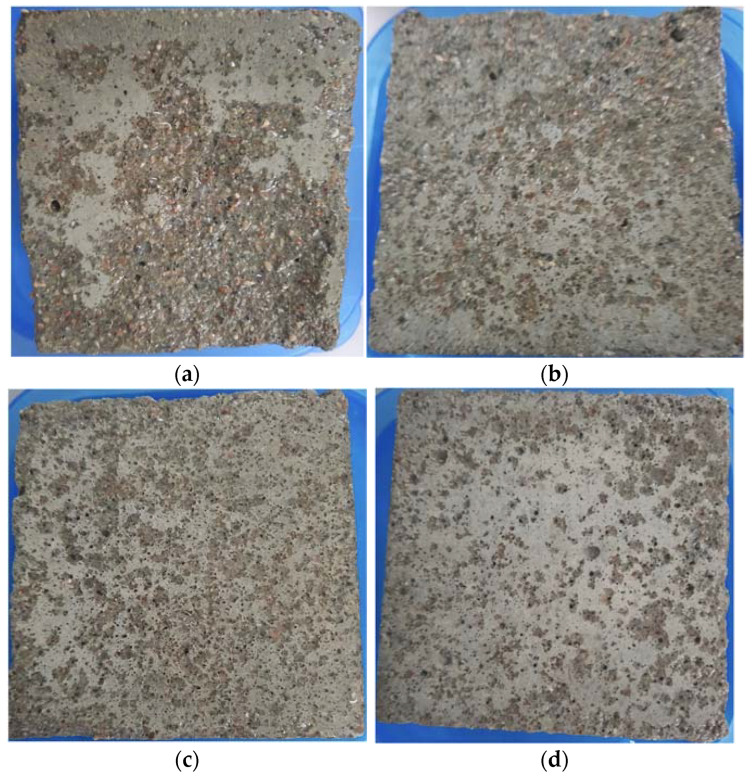
The surface of air-entraining hardened cement mortar after 14 cycles of freeze–thaw: (**a**) P3 sample; (**b**) A1 sample; (**c**) A2 sample; (**d**) A3 sample.

**Figure 19 materials-15-04382-f019:**
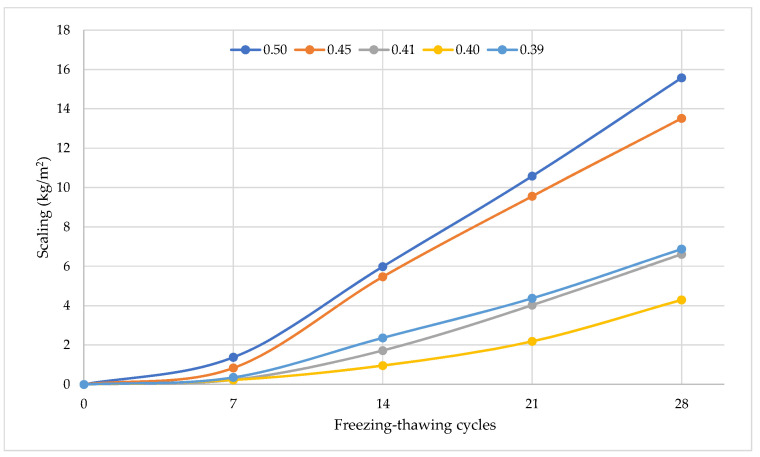
Hardened cement mortar scaling results during a freezing–thawing test in freezing water with different W/C ratio.

**Figure 20 materials-15-04382-f020:**
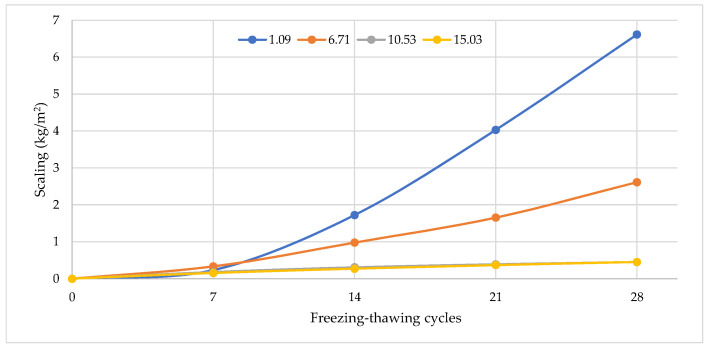
Hardened cement mortar scaling results during a freezing–thawing test in freezing water with W/C ratio and air-content.

**Figure 21 materials-15-04382-f021:**
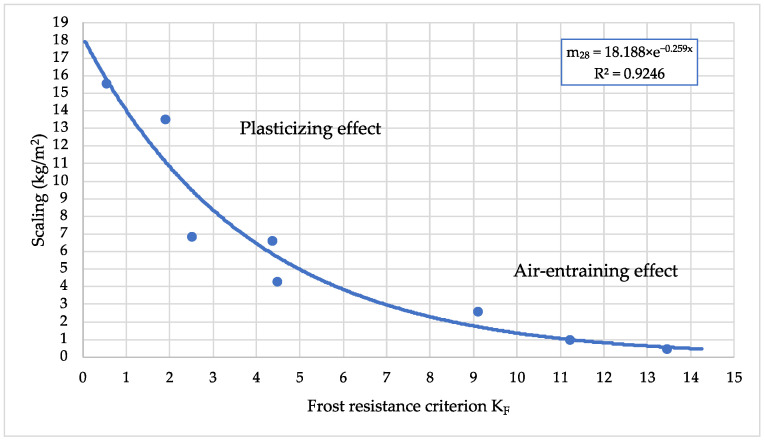
The influence of frost resistance factor on frost resistance of cement in freezing–thawing cycles.

**Table 1 materials-15-04382-t001:** Mineral and chemical composition of the cement.

Component	Amount [%]
Tricalcium silicate (C_3_S)	57.9
Dicalcium silicate (C_2_S)	15.6
Tricalcium aluminate (C_3_A)	7.5
Tetracalciumaluminoferrite (C_4_AF)	11.9

**Table 2 materials-15-04382-t002:** Chemical composition of the cement.

	Component									
	Al_2_O_3_	Fe_2_O_3_	SiO_2_	CaO	MgO	SO_3_	K_2_O	Na_2_O	Cl^-^	CaO_free_
Amount [%]	5.23	3.44	20.63	63.56	3.13	0.78	1.15	0.10	0.007	1.4

**Table 3 materials-15-04382-t003:** Physical and mechanical properties of cement.

Property	Value
2-day compressive strength, [MPa]	28 ± 2
28-day compressive strength, [MPa]	55 ± 3
Initial setting time, [min]	180
Final setting time, [min]	225
Volume stability, [mm]	1.0
Water consumption, [%]	26.6
Residue on the 90 µm sieve, [%]	1.5
Fineness by Bline, m^2^/kg	340

**Table 4 materials-15-04382-t004:** Physical and chemical properties of the used PCE.

Polymer Type	Polycarboxylate Ether (HPEG 2400)
Appearance	White solid
Hydroxyl number (mg KOH/g)	app.117
pH-value EN 1262	5.5–7.5
Molecular weight	2400
Unsaturation (%>)	95
Color (Pt-Co<)	100

**Table 5 materials-15-04382-t005:** Physical and chemical properties of the used SikaControl-10LPSA.

Polymer Type	SikaControl-10LPSA
Density	1.00 ± 0.02 kg/L;
pH value	7.0 ± 0.5
Total chlorine ion content	− <0.1% by weight of the substance
Sodium oxide equivalent	− <0.3% by weight of the substance

**Table 6 materials-15-04382-t006:** Composition and properties of the used antifoam agent.

Polymer Type	Antifoaming Agent (A-316)
Appearance	Whitish powder with good fluidity
Bulk density (kg/m^3^)	400–700
Dispersing ability in water	Easy dispersing into water
pH value of 1% in water 20 °C	7–9
Solubility	Surface treated, cold water soluble

**Table 7 materials-15-04382-t007:** Composition, consistency, and density of the mixtures with plasticizing admixtures.

№	Cement, kg	Sand, kg	Water, kg	W/C	PCE, %	AF, %	D_av_, mm	Density, kg/m^3^
C0	0.972	2.916	0.486	0.50	0	0	156	2280
P1	0.972	2.916	0.438	0.45	0.1	0.01	142	2290
P2	0.972	2.916	0.398	0.41	0.2	0.02	147	2280
P3	0.972	2.916	0.385	0.40	0.3	0.03	134	2269
P4	0.972	2.916	0.375	0.39	0.4	0.04	133	2280

**Table 8 materials-15-04382-t008:** Composition, consistency, and density of the mixture with plasticizing and air-entraining admixtures.

№	Cement, kg	Sand, kg	Water, kg	W/C	PCE, %	AF,%	AIR, %	D_av_, mm	Density, kg/m^3^
P2	0.972	2.916	0.398	0.41	0.2	0.02	0	147	2280
A1	0.972	2.916	0.398	0.41	0.2	0.02	0.1	171	2150
A2	0.972	2.916	0.398	0.41	0.2	0.02	0.2	190	2062
A3	0.972	2.916	0.398	0.41	0.2	0.02	0.3	197	1959

**Table 9 materials-15-04382-t009:** Results of freeze–thaw resistance of hardened cement paste after 28 freeze–thaw cycles.

Series	Density, kg/m^3^	Porosity, %		K_F_	Scaling after 28 Cycles, kg/m^2^
		Open	Closed		
C0	2277	14.32	0.71	0.55	15.57
P1	2281	12.71	2.18	1.91	13.51
P2	2285	10.58	4.16	4.37	6.61
P3	2271	10.88	4.38	4.47	4.29
P4	2282	9.97	2.26	2.52	6.87
A1	2159	10.69	8.75	9.09	2.61
A2	2065	11.42	11.53	11.22	0.97
A3	1972	11.95	14.47	13.45	0.45

## Data Availability

Data contained within the article.

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
