# Peer review of "Both Plasticizing and Air-Entraining Effect on Cement-Based Material Porosity and Durability"

_materials, 2022, doi:10.3390/ma15134382_

Round 1

Reviewer 1 Report

This work investigated the influence of a complex application of both plasticizing and air-entraining effects on concrete with polycarboxylate ether, air-entraining admixture, and anti-foaming agent on the air content, density, workability, freezing-thawing resistance, flexural and compressive strength of hardened cement mixture. The paper cannot be accepted in its present form as it needs further improvements. 

1. Abstract: The text must be carefully revised. Some sentences contain mistakes. In a research paper, it is expected that the introduction section briefly explains the starting background and, even more important, the originality (novelty) and relevancy of the study is well established. Once this is done, the hypothesis and objectives of the study need to be addressed, as well as a brief justification of the conducted methodology.

2. The introduction part does not have a flow or direction. It has too many different medical terminologies thrown randomly. Proper references need to be used rather than using others. Language can be improved. The sentences are half-constructed or incomplete so that the readers are expected to fend for themselves to understand their meaning.

3. Author must enrich the references with the latest developments. Some of the recent references can be added. The authors have not paid attention to previous research papers and concerns.

4. The innovation contribution of this article is not clearly stated. The research contributions should be highlighted in the revised manuscript. There is a certain lack of a clear line and message, and my strong advice to the authors would be to consider the overall structure and significantly shorten the manuscript.

5. Provide a proper reference for the equations. It is well known and available in much literature. You can consider removing a few. Please explain and define all the variables in the equations and check the manuscript thoroughly and define the variables where necessary; otherwise, readers cannot understand the equations.

6. Figures 1 and 2 are cropped from some reports. Replace with better resolution. Table 5- replace the word “soluable” with correct spelling. All figures need to be represented with bold colors (not gray). Figures 4 and 6 are very poorly represented. Figure 9 is misleading.

7. There are many linguistic and grammatical typos. Please carefully read through and conduct the proofreading.

·       Line 16-18, 20-21, 54-55, 67-68, 129, 136-137, 146, 178, 278-279 and so many: Your sentence may be unclear or hard to follow. Consider rephrasing.

8. At the end of the manuscript, please describe the scheme of the intended application of the developed method in real practice. What conditions must be met? What preliminary analysis should be carried out? What is the expected performance of this method? What are the limitations of this method?

9. Conclusions Section: Improve the conclusions section. It is very general and does not clearly explain the main objectives achieved in this research.

The list could go on, but the bottom line is that the authors need to rewrite the paper or reconsider the research content before it could be considered for publication in this journal. 

Author Response

Dear Reviewer,

We thank you for reviewing our manuscript for publication in Materials Journal. We thank you for referee report on our manuscript and would like to resubmit our revised manuscript entitled “Both plasticizing and air-entraining effect on cement-based materials porosity and durability” (Manuscript NO. 1756030) for further consideration.

We have revised the issues brought up by the editor and reviewer, the point-by-point response can be found below. The changes in the manuscript are highlighted in red.

Sincerely,

Aigerim Tolegenova,
Satbayev University, Satpayev str.22a, 050013 Almaty, Kazakhstan. [email protected]

According to the list, we have made the following changes:

Response to Reviewer 1 Comments

Point 1: Abstract: The text must be carefully revised. Some sentences contain mistakes. In a research paper, it is expected that the introduction section briefly explains the starting background and, even more important, the originality (novelty) and relevancy of the study is well established. Once this is done, the hypothesis and objectives of the study need to be addressed, as well as a brief justification of the conducted methodology.

Response 1: The whole article's abstract has been revised and corrections have been made to the text. New abstract was involved in the article.

Point 2: The introduction section does not have a flow or direction. It has too many different medical terminologies thrown randomly. Proper references need to be used rather than using others. Language can be improved. The sentences are half-constructed or incomplete so that the readers are expected to fend for themselves to understand their meaning.

Response 2: We have rephrased and changed the direction in introduction section. The sentences have been correctly constructed. Language was improved.

Point 3: Author must enrich the references with the latest developments. Some of the recent references can be added. The authors have not paid attention to previous research papers and concerns.

Response 3: The list of references (36-48) was developed with the latest publications.

Point 4: The innovation contribution of this article is not clearly stated. The research contributions should be highlighted in the revised manuscript. There is a certain lack of a clear line and message, and my strong advice to the authors would be to consider the overall structure and significantly shorten the manuscript.

Response 4: In the end of Introduction section we included information about innovation contribution of the article. The innovation contribution is explained as combined effect of plasticizing and air-entraining admixtures on the freezing-thawing resistance of hardened cement and this effect analysis during concrete mixture proportioning. The methodological section of manuscript was significant shortened.

Point 5: Provide a proper reference for the equations. It is well known and available in much literature. You can consider removing a few. Please explain and define all the variables in the equations and check the manuscript thoroughly and define the variables where necessary; otherwise, readers cannot understand the equations. 

Response 5: We considered removing some formulas, as they are available in much literature.

Point 6: Figures 1 and 2 are cropped from some reports. Replace with better resolution. Table 5- replace the word “soluable” with correct spelling. All figures need to be represented with bold colors (not gray). Figures 4 and 6 are very poorly represented. Figure 9 is misleading. 

Response 6: Figure 1 and Figure 2 were corrected. The word "soluable"  has been replaced by "solubility". All Figures have been revised with bold colors. The equation in Figure 9 has been corrected as well.

Point 7: There are many linguistic and grammatical typos. Please carefully read through and conduct the proofreading.

Response 7: The sentences from 16-18, 20-21, 54-55, 67-68, 129, 136-137, 146, 178, 278-279 line were corrected as well whole manuscript.

Point 8: At the end of the manuscript, please describe the scheme of the intended application of the developed method in real practice. What conditions must be met? What preliminary analysis should be carried out? What is the expected performance of this method? What are the limitations of this method?

Response 8: After conclusion we added information about intended application of results in the practice and limitations of this method.

Point 9: Conclusions Section: Improve the conclusions section. It is very general and does not clearly explain the main objectives achieved in this research.

Response 9: We have corrected the conclusion as suggested.

Reviewer 2 Report

The manuscript focuses on plasticizing and air-entraining effect on cement-based materials porosity and durability, which fit with the journal's aims and scope. I think it can be considered for possible publication in the journal after a minor revision. Some recommendations are as following:

1. In Figure 9, R2=0.001, please explain.

2. The format of references are inconsistent, please revise.

3. What is the curing condition in chapter 2.2, and why choose 14 days curing age?

4. Please discuss the influence of cement content and curing age on porosity and durability of HCM.

5. The literature review should improve with the addition of the following works: 10.1016/j.jclepro.2022.131527, 10.1007/s11356-021-15769-3, 10.3390/polym14101995

6. Conclusions are lengthy. Some information's can be moved to the result and discussion section.

Author Response

Dear Reviewer,

We thank you for reviewing our manuscript for publication in Materials Journal. We thank you for referee report on our manuscript and would like to resubmit our revised manuscript entitled “Both plasticizing and air-entraining effect on cement-based materials porosity and durability” (Manuscript NO. 1756030) for further consideration.

We have revised the issues brought up by the editor and reviewer, the point-by-point response can be found below. The changes in the manuscript are highlighted in red.

Sincerely,

Aigerim Tolegenova,
Satbayev University, Satpayev str.22a, 050013 Almaty, Kazakhstan. [email protected]

Response to Reviewer 2 Comments

Point 1: In Figure 9, R2=0.001, please explain.

Response 1: We have revised the R indicator in the article from experimental results.

Point 2: The format of references are inconsistent, please revise.

Response 2: We have corrected the references as suggested.

Point 3: What is the curing condition in chapter 2.2, and why choose 14 days curing age?

Response 3: Mortar specimens curing was carried out in water at 20±2°C. 14 days of curing were used to achieve sufficient mass losses during cyclic freezing-thawing of hardened concrete with air-entraining admixtures.

Point 4: Please discuss the influence of cement content and curing age on porosity and durability of HCM.

Response 4: We have not tested the influence of cement content and curing age on the porosity and durability of HCM. These parameters should not affect the trends of the issues considered in our article (influence of W/C ratio and entrained air content). With increasing hardening time, capillary porosity will decrease and frost resistance will increase. Increase in the amount of cement with the same amount of water, capillary porosity decreases, and frost resistance increases.

Point 5: The literature review should improve with the addition of the following works: 10.1016/j.jclepro.2022.131527, 10.1007/s11356-021-15769-3, 10.3390/polym14101995

Response 5: The works with DOI: 10.1016/j.jclepro.2022.131527, 10.1007/s11356-021-15769-3 have been included in Introduction section of article and we considered that article 10.3390/polym14101995 does not address issues related to our research.

Point 6: Conclusions are lengthy. Some information's can be moved to the result and discussion section.

Response 6: We have shortened the Conclusions section and moved some information to Discussion section.

Reviewer 3 Report

The article considers the influence of superplasticizers and air-entraining admixtures on the technological properties of hardened cement mortar. The authors have done a great work and presented a good article. The article is well and clearly written, reveals in detail the progress of the work and the obtained results. The results may be interesting for readers of Materials and useful for practical application. I believe the article can be published in Materials, but I recommend that the authors pay attention to the following comments before accepting the article for publication.

1) Line 36:  W/C needs to be revealed for the first time.

2) At the beginning of Section 2 Materials and Methods, it is recommended to provide a brief rationale for the materials and procedures described below.

3) Fig. 2. Check the captions in the figure.

4) Why were prisms with a size of 40 mm×40 mm×160 mm chosen for compression tests?

5) Please check the coefficient of determination in figure 9.

6) Discussion of the results and their comparison with the results of other publications is necessary.

Author Response

Dear Reviewer,

We thank you for reviewing our manuscript for publication in Materials Journal. We thank you for referee report on our manuscript and would like to resubmit our revised manuscript entitled “Both plasticizing and air-entraining effect on cement-based materials porosity and durability” (Manuscript NO. 1756030) for further consideration.

We have revised the issues brought up by the editor and reviewer, the point-by-point response can be found below. The changes in the manuscript are highlighted in red.

Sincerely,

Aigerim Tolegenova,
Satbayev University, Satpayev str.22a, 050013 Almaty, Kazakhstan. [email protected]

Response to Reviewer 3 Comments

Point 1: Line 36:  W/C needs to be revealed for the first time.

Response 1: In the 36 line Introduction section, we did not consider the specific values of the W/C indicator. In our studies, W/C was reduced the from 0.5 to 0.38 due to plasticizing effect of admixture.

Point 2: At the beginning of Section 2 Materials and Methods, it is recommended to provide a brief rationale for the materials and procedures described below.    

Response 2: We have corrected Materials and Methods section as suggested.

Point 3: Fig. 2. Check the captions in the figure.

Response 3: The captions in Figure 2 have been corrected.

Point 4: Why were prisms with a size of 40 mm×40 mm×160 mm chosen for compression tests?

Response 4: According to the European standard, beams of size 40x40x160 mm are used to study standard cement mortars for flexural and compression strength.

Point 5: Please check the coefficient of determination in figure 9.

Response 5: We have revised and changed the coefficient in Figure 9.

Point 6: Discussion of the results and their comparison with the results of other publications is necessary.

Response 6: We have discussed and compared our results of research with over 10 references. They are represented by numbers 36-48 in the list of references.

Reviewer 4 Report

This article studies the influence of different chemicals on the strength of concrete. Although a great deal of work has been done, there are still many unclear points about the specific means and the basis of the research means in the research process. In the end, the article doesn't link the research content with the actual work, but only stays in the theoretical aspect, and the research in this area is relatively mature at present. In addition, the article does not introduce the manufacturing process of the test block and the physical and mechanical experiments in detail. The discussion part only describes the data obtained, but does not make quantitative analysis in the deep level. Therefore, a major revision is needed to realize the real value of the research results. The following issues also need to be improved:

      The abstract should briefly describe the validity of publications, including the methods, analyses and results of the obtained research.

      The paper should clearly emphasize what is the purpose and scope of the work.

      Please, develop the description of the methodology.

      The authors could add a paragraph with a brief description of the extent to which the presented research and results contribute to science.

Author Response

Dear Reviewer,

We thank you for reviewing our manuscript for publication in Materials Journal. We thank you for referee report on our manuscript and would like to resubmit our revised manuscript entitled “Both plasticizing and air-entraining effect on cement-based materials porosity and durability” (Manuscript NO. 1756030) for further consideration.

We have revised the issues brought up by the editor and reviewer, the point-by-point response can be found below. The changes in the manuscript are highlighted in red.

Sincerely,

Aigerim Tolegenova,
Satbayev University, Satpayev str.22a, 050013 Almaty, Kazakhstan. [email protected]

Response to Reviewer 4 Comments

Point 1: The abstract should briefly describe the validity of publications, including the methods, analyses and results of the obtained research.

Response 1: We have corrected the abstract as suggested.

Point 2: The paper should clearly emphasize what is the purpose and scope of the work.

Response 2: Apologies for it being missing from the original manuscript. We have added a purpose and scope of the work in revised manuscript.

Point 3: Please, develop the description of the methodology.

Response 3: We have revised the methodology section.

Point 4: The authors could add a paragraph with a brief description of the extent to which the presented research and results contribute to science.

Response 4: At the end of Introduction section we have included a brief description of our research contribution to science.